# Searching for biological feedstock material: 3D printing of wood particles from house borer and drywood termite frass

**Rudy Plarre[1], Andrea Zocca[1], Andrea Spitzer[1], Sigrid Benemann[1], Anna A. Gorbushina[1,2], Yuexuan Li[1,3], Anja Waske[1], Alexander Funk[1], Janka Wilbig[1], Jens Günster**[1,4]*

**1** Bundesanstalt für Materialforschung und -prüfung BAM, Berlin, Germany, **2** Institute of Geological Sciences, Freie Universität Berlin, Berlin Germany, **3** State Key Laboratory for Manufacturing Systems Engineering, School of Mechanical Engineering, Xi'an Jiaotong University, Xi'an, Shaanxi, China, **4** Institute of Non-Metallic Materials, TU Clausthal, Clausthal-Zellerfeld, Germany

* jens.guenster@bam.de

**Data Availability Statement:** All relevant data are within the manuscript and its Supporting Information files.

**Funding:** The author(s) received no specific funding for this work.

## Abstract

Frass (fine powdery refuse or fragile perforated wood produced by the activity of boring insects) of larvae of the European house borer (EHB) and of drywood termites was tested as a natural and novel feedstock for 3D-printing of wood-based materials. Small particles produced by the drywood termite Incisitermes marginipennis and the EHB Hylotrupes bajulus during feeding in construction timber, were used. Frass is a powdery material of particularly consistent quality that is essentially biologically processed wood mixed with debris of wood and faeces. The filigree-like particles flow easily permitting the build-up of wood-based structures in a layer wise fashion using the Binder Jetting printing process. The quality of powders produced by different insect species was compared along with the processing steps and properties of the printed parts. Drywood termite frass with a Hausner Ratio HR = 1.1 with $\rho$Bulk = 0.67 g/cm$^3$ and $\rho$Tap = 0.74 g/cm$^3$ was perfectly suited to deposition of uniformly packed layers in 3D printing. We suggest that a variety of naturally available feedstocks could be used in environmentally responsible approaches to scientific material sciences/additive manufacturing.

## Introduction

Anthropogenic perturbations of natural ecosystems are omnipresent: materials and products of human activity are superimposed on natural cycles everywhere. According to Schellnhuber 1999 [1] there are two main components: the ecosphere N and the human factor H. N consists of intricate linkages between the atmosphere, hydrosphere, cryosphere, lithosphere, biosphere, etc, while the human factor H aggregates all actions and products along with a metaphysical component of human activity.

Sustainable coevolution of the ecosphere and the anthroposphere requires fresh scientific attitudes and approaches, including completely new ways of manufacturing. The ever-increasing human impact on the planet requires the deliberate coupling of natural feedstocks to novel

**Competing interests:** The authors have declared that no competing interests exist.

manufacturing process. This way the life cycle of the products and materials can be determined early in production. Substituting dedicated feedstocks for additive manufacturing (AM) with surplus natural materials is one way to substantially increase AM sustainability while concomitantly providing high-value outputs for "pre-owned" materials/products.

As a general rule for all production processes, natural and recycled feedstocks should take preference to dedicated ones–especially in the context of a circular economy. Deliberately developing naturally available feedstock constitutes an environmentally responsible scientific approach in material sciences.

## Additive manufacturing

Adding material to form an object instead of subtracting material from an excessively large block is a new trend in manufacturing technologies which is currently stimulating an entire industry [2]. In the majority of additive manufacturing processes, the material is added layer by layer. The raw material (feedstock) is fed into the process as a powder/granulate, paste or suspension, as it is in a state optimized for the layer deposition process. In the manufacturing process itself, the feedstock is used to build up the desired object and it is simultaneously transferred into a state possessing its final physical properties, or at least providing enough mechanical strength to transfer the configured object to further processing steps. Adding instead of subtracting material implies more than just flexibility in design. Multi material processing, the generation of unique properties and functionalities and functionally graded materials are just some facets intrinsic to AM. One of the most popular and widespread additive manufacturing technologies, the "Binder Jetting" (BJ), is making use of a powdery material as feedstock [3, 4], see also Fig 1. A layer of powdered material is first spread as a layer and subsequently the corresponding layer information of the object manufactured is selectively inscribed by a printing head, spraying individual droplets of a binding liquid onto the powder layer, thus selectively consolidating the powder and defining the cross-section of the object in a respective layer.

The popularity of BJ is based on the fact, that it virtually can accept all powdery materials which provide sufficient flowability to be spread as a homogeneous thin layer. This flexibility has stimulated the creativity of many research groups envisioning the use of abundant materials, such as sand [5–7], or recyclable material in powdery form for upgrading waste for the manufacture of new products [8]. Costs and resources for the initial material synthesis can be saved. However, the refinement of powdery raw materials to powders well suited for BJ remains a mandatory processing step if the recycled material is not directly obtained in the appropriate powdery state [9].

Here a strategy of coupling the development of the most promising AM approaches to considerations of the use of natural or Nature-recycled materials as feedstocks was applied. In the present work the powdery material which remains when house borer larvae or drywood termites feed on wood, the so called frass, is used as a novel feedstock. 3D printing of wood chops [10–14] or plastic-wood composites has already been proven to be a feasible way for obtaining objects with wooden haptics. Dedicated feedstocks have been processed from wood and have been adapted to the respective printing process by refining it with polymeric additives [15–18]. In terms of sustainability, 3D printing of wood-based materials from house borer and termite frass is going one step further as it is not only using naturally occurring but also naturally processed materials directly as feedstocks.

In the BJ process, the binder system, which is used to consolidate the powdery material to form an object, has to fulfill multiple requirements. As a liquid, it has to be of an appropriate viscosity and surface tension to be dosed by a commercial printer head [4, 19, 20]. In order to penetrate the deposited layer, it must also moisten the powdery feedstock. Moreover, it ideally

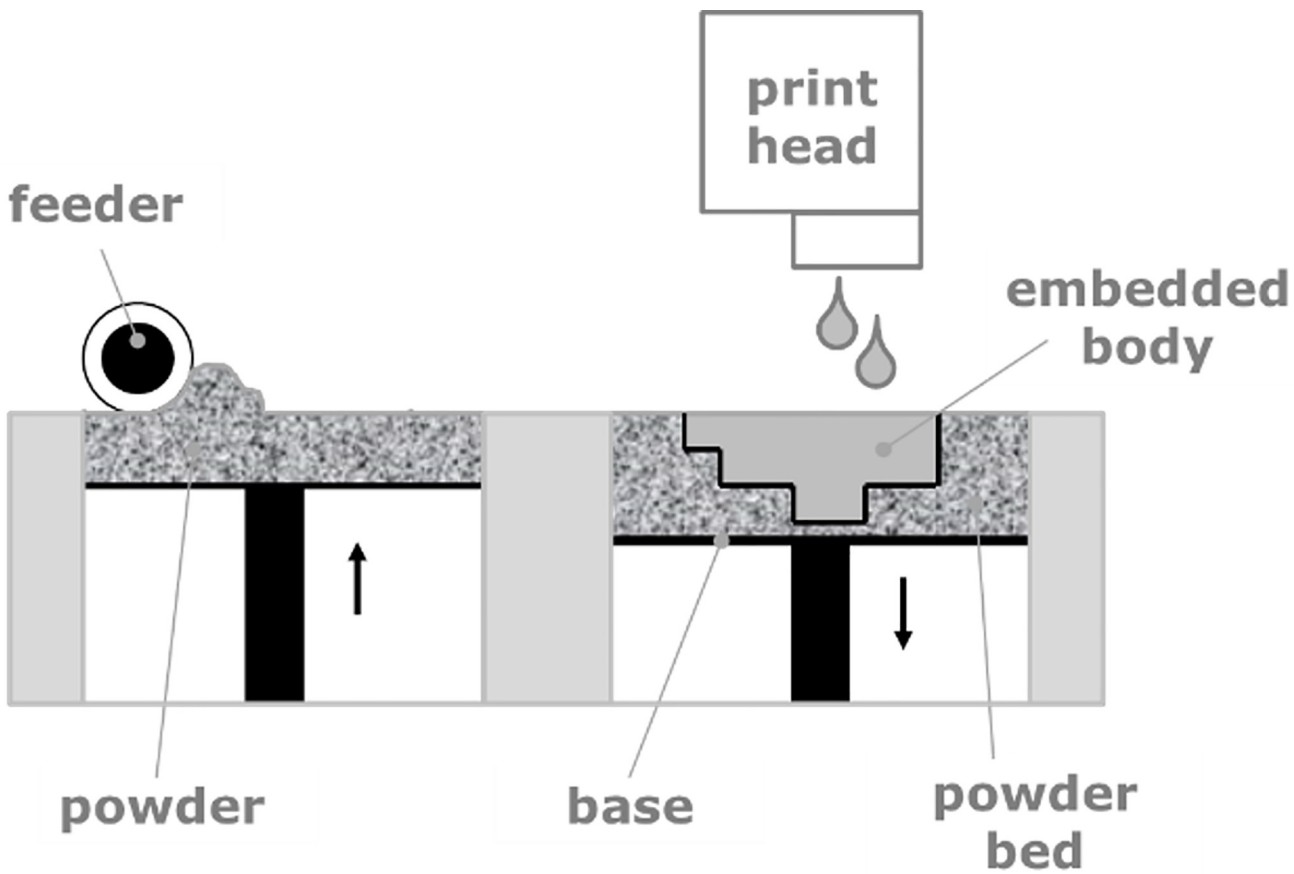

**Fig 1. Schematic of the Binder Jetting (BJ) process.**

interacts with the powdery material to form a strong interparticle adhesion. In most cases, addition of a binder does not result in a significant densification of the powder. Hence, BJ is typically providing porous parts requiring a post treatment for densification [21, 22].

## Feedstocks from timber

A major interest exists in 3D printing for ecofriendly and recycled materials, particularly in Binder Jetting [11–16, 23, 24]. BJ can utilize basically anything that can be powdered to an appropriate particle size. The particle size of the powders is essential for obtaining sufficient flowability for the deposition of defect-free layers: In case of too fine powders, the flowability will be poor, in case of too coarse-grained powders the definition of the part will be imprecise [25]. Wood particles may be obtained as byproducts from wood machining, such as saw dust, or are deliberately processed from wood. For obtaining suitable feedstocks for 3D printing, in most published works the wood particles are mixed into a polymer or mixed with other binding phases [15, 17, 26–29]. On the other hand, timber can also be naturally processed into printable powders by e. g. insects feeding on wood. We have used small particles from feeding byproducts of the European house borer Hylotrupes bajulus and drywood termite Incisitermes marginipennis as raw material for 3D-printing.

Insects in construction timber share several anatomic and physiological features which make them to appear perfectly adapted to this environment. Wood is an inhomogeneous poriferous matrix containing mainly cellulose and lignin. Their relative amounts vary between

heartwood and sapwood as well as in the early and late wood of the annual rings. This results in local strength differences and an uneven distribution of essential nutrients for the insects. The larvae of wood boring beetles or drywood termites have strong mandibles which allow abrasion of all parts. The cellulose is the main hydrocarbon source usually digested with the aid of cellulase-producing microorganisms. However, the rare nitrogen containing elements of wood are the limiting factors. In order to access as much nitrogen as possible much more wood is consumed by wood feeding insects than actually needed for development. Larvae of H. bajulus, e. g. excavate throughout the sapwood leaving extensive tunnels filled with frass. The frass contains either loosely chopped off wood particle (debris) or dense-packed faeces. The latter is of cylindrical shape made up out of himidigested cellulose/lignin conglomerates. Destructions of insects feeding on build-in timber can cause severe danger and precautions are needed. Laboratories like BAM therefore rear large pest populations to test different control strategies for efficacy evaluation in pest control. The rearing byproducts like the frass were of no further use and discarded. However, after being modified by the insect digestive system the former non-uniform wooden material is changed to a homogeneous compact cellulose-lignin mixture and becomes suitable for further technical applications, such as 3D printing, without any further processing. In the present study we have evaluated, the so called frass, as feedstocks for 3D printing. The morphology of the drywood termite frass is quite different to house borer frass. While drywood termite frass appears as six-sided pellets almost uniform in size the frass from house borer is sawdust-like and more irregularly shaped. In contrary to drywood termites, the house borer larvae digest only part of the abraded wood with their frass containing loosely chopped off debris as well as more dense-packed faeces.

## Materials and methods

Larvae of European house borer (EHB) were reared at constant conditions of $28 \pm 2°C$ and $75 \pm 5\%$ r.h. During the first days after hatching from the eggs, larvae were manually inserted into pine, Pinus sylvestris, sapwood blocks ($1.5 \times 2.5 \times 5\ cm^3$) enriched with peptone and yeast. This enrichment with nutrients was carried out by impregnating the sapwood with an aqueous solution of 1% peptone and 0.3% yeast at low pressure of 100 to 200 mbar for 30 minutes to speed up development. Two larvae per wood block were allowed to feed for approximately six months before being individually transferred into pure pine sapwood blocks ($3 \times 4 \times 5.5\ cm^3$), not enriched with any nutrients. While feeding in the wood, the larvae produce debris and faeces which are left behind in the frass tunnels. Debris is undigested wooden material derived from abrasion processes when the larvae's mandibles carve on the wood. It usually bypasses the larvae during movement through the wood. While debris are of undefined structure, the faeces are densely packed into cylindrical pellets when leaving the larva's hindgut. As the wood is increasingly consumed by the larvae, frass (debris and faeces) eventually trickle out and can be collected in larger amounts.

Using a vibratory sieve shaker (Analysettre 3 spartan, Fritsch, 55743 Idar-Oberstein, Germany) EHB frass was sieved for 30 min at 1 mm amplitude. A fraction with particle size distribution of 45 μm– 100 μm, which amounts to 17% of the total frass, 57% above 100 μm and 26% below 45 μm, was then used for 3D-printing, see also Fig 2. The frass particles show a considerable flowability although the particle shape is rather flake-like than spherical. The Hausner ratio (HR), as a measure of flowability [29], was determined according to the relation shown in Eq 1:

$$\rho Tap / \rho Bulk = HR \qquad (1)$$

where HR is the Hausner ratio, ρBulk displays the freely settled bulk density of the powder and

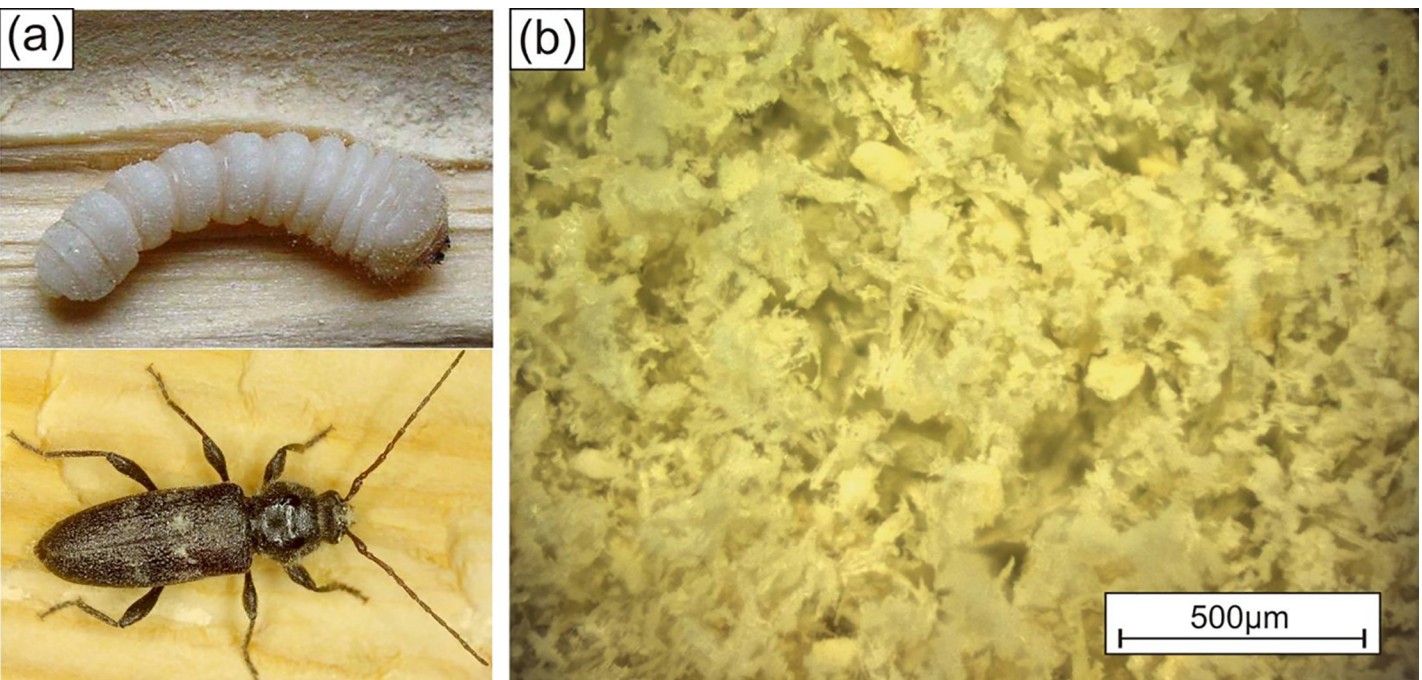

**Fig 2.** a) European house borer (Hylotrupes bajulus) full grown larva (top) and adult beetle (bottom); b) the sieved frass (in a particle size fraction of 45 to 100 μm) produced by larvae and used for 3D-printing.

ρTap the bulk density after a given number of tapping cycles, at which the bulk density is in a plateau, in g/cm$^3$. The tapped density was determined according to ISO 787–11 with a STAV 2003 type equipment from J. Engelsmann AG, Germany. With ρBulk = 0,14 g/cm$^3$ and ρTap = 0,18 g/cm$^3$, the EHB frass particles have a Hausner ratio of 1.25 which displays a fair flowability according to the classification [30].

Within this study, frass from drywood termites has been considered as feedstock for 3D printing, as well, see also Fig 3. Termites rely on fungi, protists and bacteria that live in their

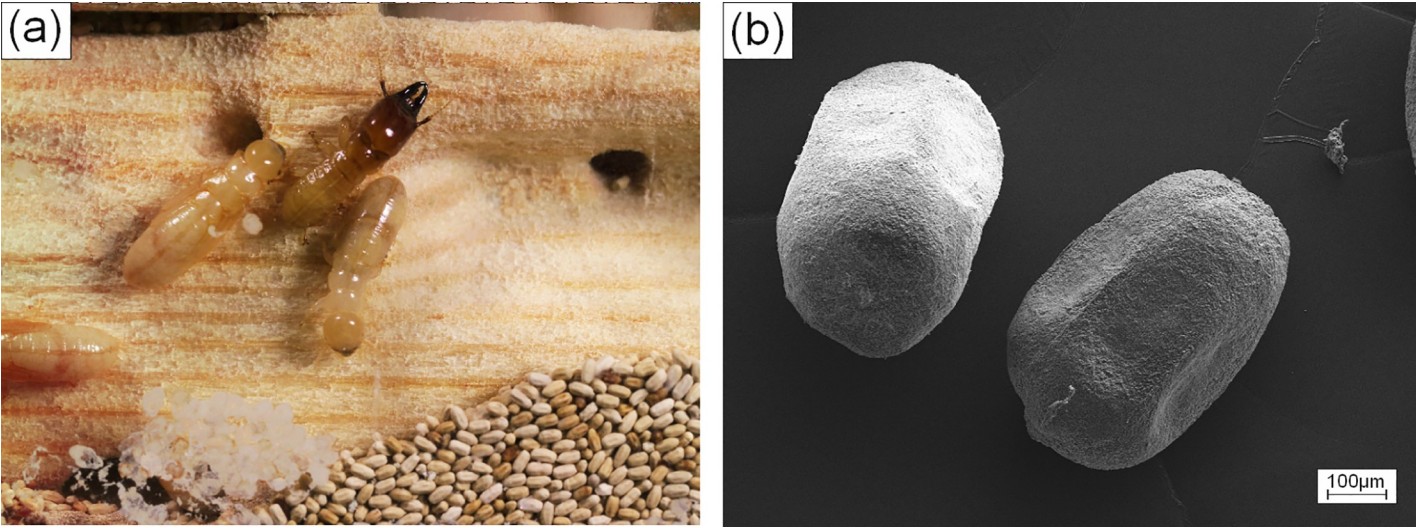

**Fig 3.** a) Drywood termites (Incisitermes marginipennis), soldier and worker with frass; b) SEM micrograph of single pellets of drywood termite frass.

gut to break down the wood and digest lignin and cellulose. In comparison to the feeding byproducts of EHB the drywood frass contains six-sided pellets almost uniform in size and reveal an excellent flowability required for the layer wise buildup of wooden structures, see also Fig 3. The pellets are very compact and composed of fine fibers and particles. Their good flowability, HR = 1.1 with $\rho$Bulk = 0,67 g/cm³ and $\rho$Tap = 0.74 g/cm³, making them perfectly suited for the deposition of uniformly packed layers in 3D printing.

The termite pellets have been imaged by 3D X-ray computed tomography using a commercial ZEISS Xradia 620 Versa X-ray microscope. An acceleration voltage of 80 kV and power of 10 W was used. The X-ray spectrum was filtered on the source side by using a device specific LE1 filter. A geometrical magnification of around 6.8x (with a source to object distance of 25 mm and object to 145 mm) and optical magnification of 0.4x results in an effective pixel size of 10 µm. Since the pellets are light and experience some degree of electrostatic repulsion, they have been fixed on adhesive tape that has been rolled up for the measurement, see also Fig 4. The 3D tomographic dataset with 801 angular object projections was reconstructed using ZEISS reconstructor software, the processing of the reconstructed data was carried out using the software package Avizo (Thermo Fisher Scientific, USA). Length, Breadth and Width are ferret diameters of a measured 3D object. The length and width are the longest and shortest ferret diameters of an object, respectively. The breadth of the object was measured to be the longest ferret diameter orthogonal to the object's length.

Fig 4 shows the 3D representation of the tomographic dataset of the termite pellets. The rendered image of the pallets demonstrates that the pellets are of uniform, approximately ellipsoidal shape and have a narrow size distribution. The facetted 6-fold symmetry of the pellets,

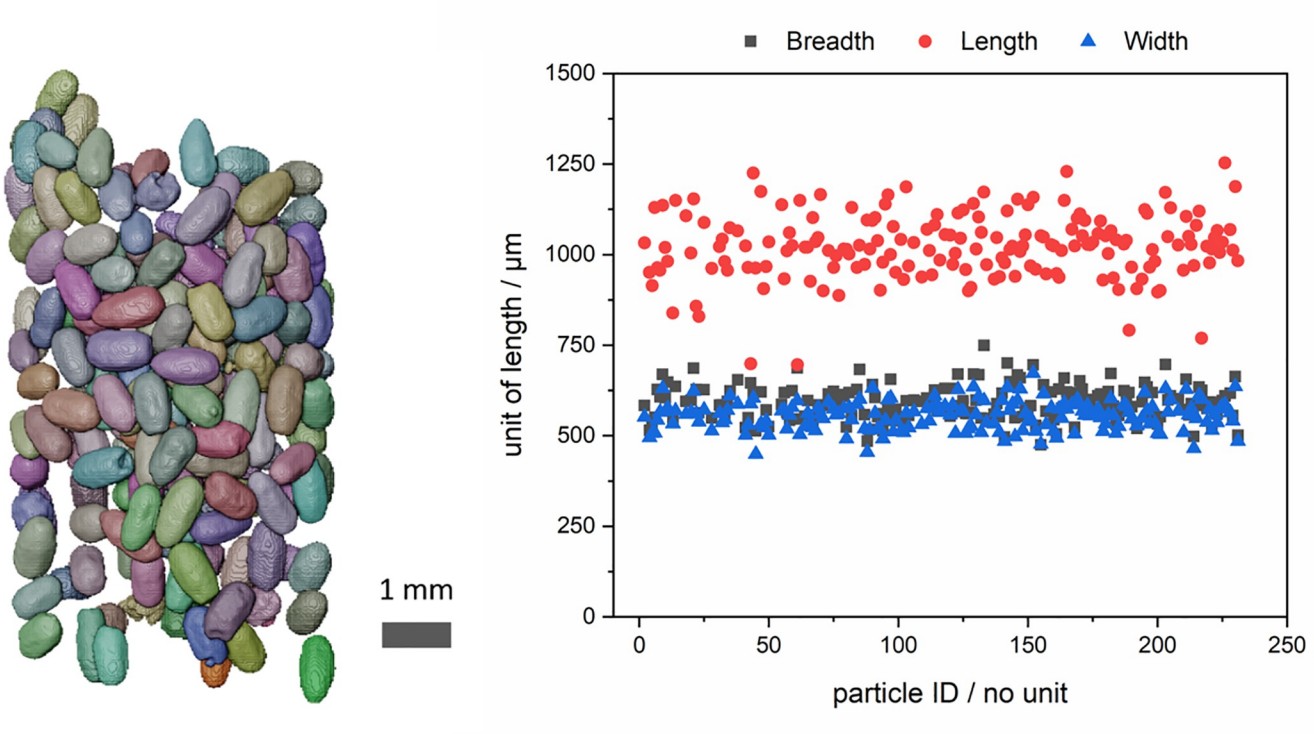

**Fig 4. Rendered X-ray tomographic dataset of the termite pellets studied, 180 pellets (left).** The length, breadth and width of each particle as determined from the 3D data (right). The average length of the pellets is around 1050 µm, its breath (or width) around 580 µm, yielding an aspect ratio of 1.8.

seen in the SEM image of Fig 3, is less obvious to observe from the 3D rendered image. The measurements of the geometrical dimensions of each of the 180 pellets are plotted in the graph of Fig 4, also. Length and breadth/width differ significantly and are around 1050 μm and 580 μm, respectively. The similarity between width and breadth indicates a well-rounded shape. The size of the individual pellets is large for BJ. In order to adapt the printing process to the size of the pellets, layer thicknesses of minimum 600 μm have been evaluated and 800 μm was found optimal for the deposition of smooth layers.

Using a commercial 3D printing machine (RX-1, Prometal RCT GmbH, Augsburg, Germany) specimens from European house borer and drywood termites were printed. The binder used was a water based commercial system provided by ExOne GmbH, Augsburg Germany (PM-B-SR2-02, viscosity of 10.7 mPa·s @ 1000s-1). Cross-linking of the binder was carried out after each layer printing in the thermal curing station of the printer. The precision of the printed geometries depend on several processing parameters (e. g. powder particle size, flow-ability, and layer thickness) as well as binder saturation. With fine standard powders, with a typical particle size < 60 μm, the printer can achieve a volumetric resolution of better than 100 μm in all three dimensions. For adapting the printer to the new powders, printing parameters such as binder saturation, layer thickness and curing time were varied. The binder saturation is a parameter useful to evaluate the amount of binder used to glue a certain quantity of powder, because it gives the ratio between the volume of binder spread out in a volumetric unit (voxel) and the free volume, not filled with powder, in the same voxel. Consequently, a saturation S of 100% corresponds to all porosity of the powder bed is filled by binder. A saturation of 100% was chosen for printing EHB frass and 166% for termite frass, respectively. The layer thickness which turned out to be most appropriate for a reproducible deposition of uniform layers was 100 μm for EHB and 800 μm for termite frass.

Fig 5 is showing the models of the structures printed. The cube in Fig 5A, with dimensions of 9 mm³ and rectangular struts, was chosen to evaluate the printing accuracy especially for

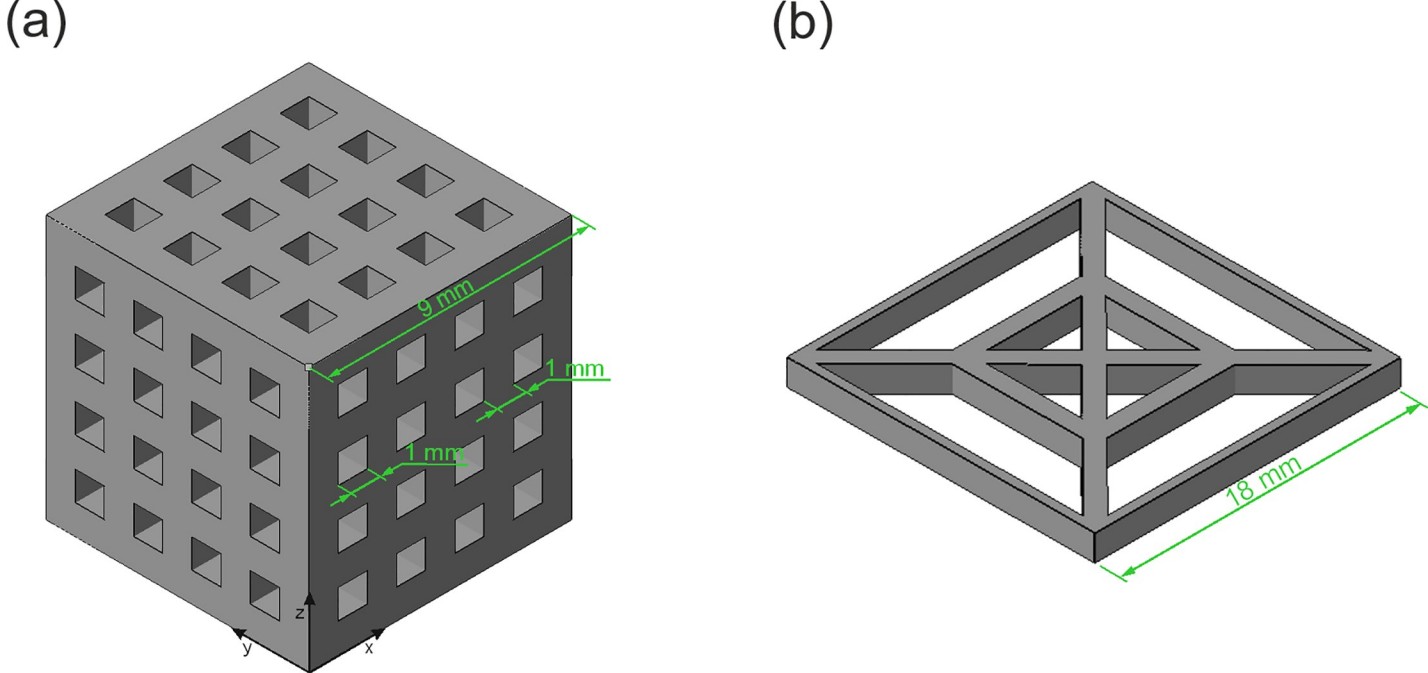

**Fig 5. Schematic drawing, including dimensions, of the specimens printed.**

the fine powders from ESB. The model in Fig 5B is designed for getting an impression about the possibility to reproduce fine structural features with particles as large as 1050 μm. It consists of an inner frame with 2 mm open space and an outer frame with 4 mm open space.

## Results and discussion

### Powder-based 3D-printing by Binder Jetting (BJ) of EHB frass

A cubic structure was printed according to the model from Fig 5A, see Fig 6A. The model structure is reproduced very well during the printing process except for the bottom plane. Clearly a convex instead of planar shape can be recognized. This distortion arises from an oversaturation with binder. Also, indicative for an oversaturation is the reduction in the diameter of the printed capillaries. Instead of the designed square shaped cross section of 1mm edge length, the edge length is reduced to approximately 500 μm. The dimensional tolerance of printed parts associated to the printing parameters chosen is in the range of 200 μm.

Due to the low packing density of the ESB frass, parts obtained from 3D printing have shown low mechanical strength and were considered as not useful in the as-printed state. On the other hand, they show considerably well detail quality and could serve as preforms for further processing steps, such as infiltration, giving them more mechanical strength. The frass particle were hardly compacted during layer deposition. Addition of binder did not result in a significant densification. Fig 6B shows details of the fluffy structure of the as-printed part.

### Powder-based 3D-printing by Binder Jetting (BJ) of drywood termite frass

From Fig 4 an average 580 μm breadth/width and 1050 μm length of the particles is deduced. This means that the approximately spheroidal particles have an aspect ratio of around 1.8. Overall, the 3D analysis reveals that termite pellets have a narrow size distribution and ellipsoidal shape. Accordingly, frass from dry wood termites reveals an excellent flowability and, thus, relatively high packing densities could be obtained during layer deposition. The approximately four times higher packing density, as compared to the EHB frass, appears promising for even load bearing applications of printed parts, without further processing.

One general problem with compacted porous particles, i.e., the frass pellets, however, exists in the BJ process: Driven by capillary forces, the binder is drawn into the fine network of pores within the individual pellets. Primarily, all capillaries within the pellets must be filled before a sufficient amount of binder remains available for gluing the pellets to each other effectively. In the competition among capillaries with smaller diameters within individual pellets and

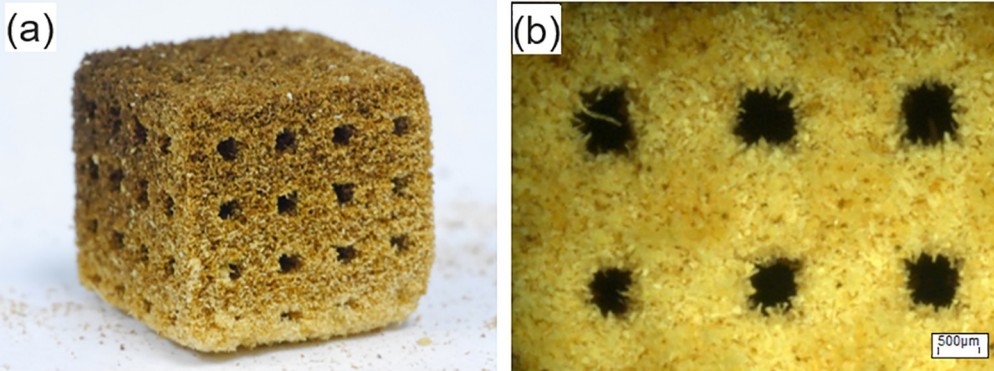

**Fig 6. Images of the manufactured specimens from European house borer (EHB) frass.** a) Photograph of a printed cubic sample. b) Light microscope image of the sample from a), showing the wooden chops and macroscopic channels.

capillaries with larger diameters between individual pellets, the smaller diameters are providing a stronger drag force for the binder uptake. Hence, a lot of binder is consumed before excessive binder is available for interparticle gluing. Therefore, in the printing process a high binder saturation of the powder bed is required for consolidating a structure while an excessive amount of binder stays in the pellets. Upon binder uptake of the frass particles, no swelling and loss of structural integrity could be observed. In order to compensate the binder uptake of the single pellets a binder saturation of 166% was chosen. It was found difficult to deposit layers on top of the first printed cross sections, due to adhesion of the binder saturated pellets to the recoater. This problem could be solved by applying a gas flow throughout the powder bed, providing an additional force stabilizing already deposited material. This technology has been introduced recently for the deposition of powders with poor flowability [31]. Fig 7 shows parts printed from termite frass according to the models shown in Fig 5.

In order to get an impression about the possibility to reproduce fine structural features in BJ with particles as large as 1050 μm, the model structure from Fig 5B has been printed, see Fig 7C. It can be seen from Fig 7C that the inner square frame of the model with ca. 2 mm open space is not well reproduced, while the outer, with ca. 4 mm open space, is reasonably well resolved. Accordingly, the dimensional tolerance of printed parts is in the range of 1000 μm.

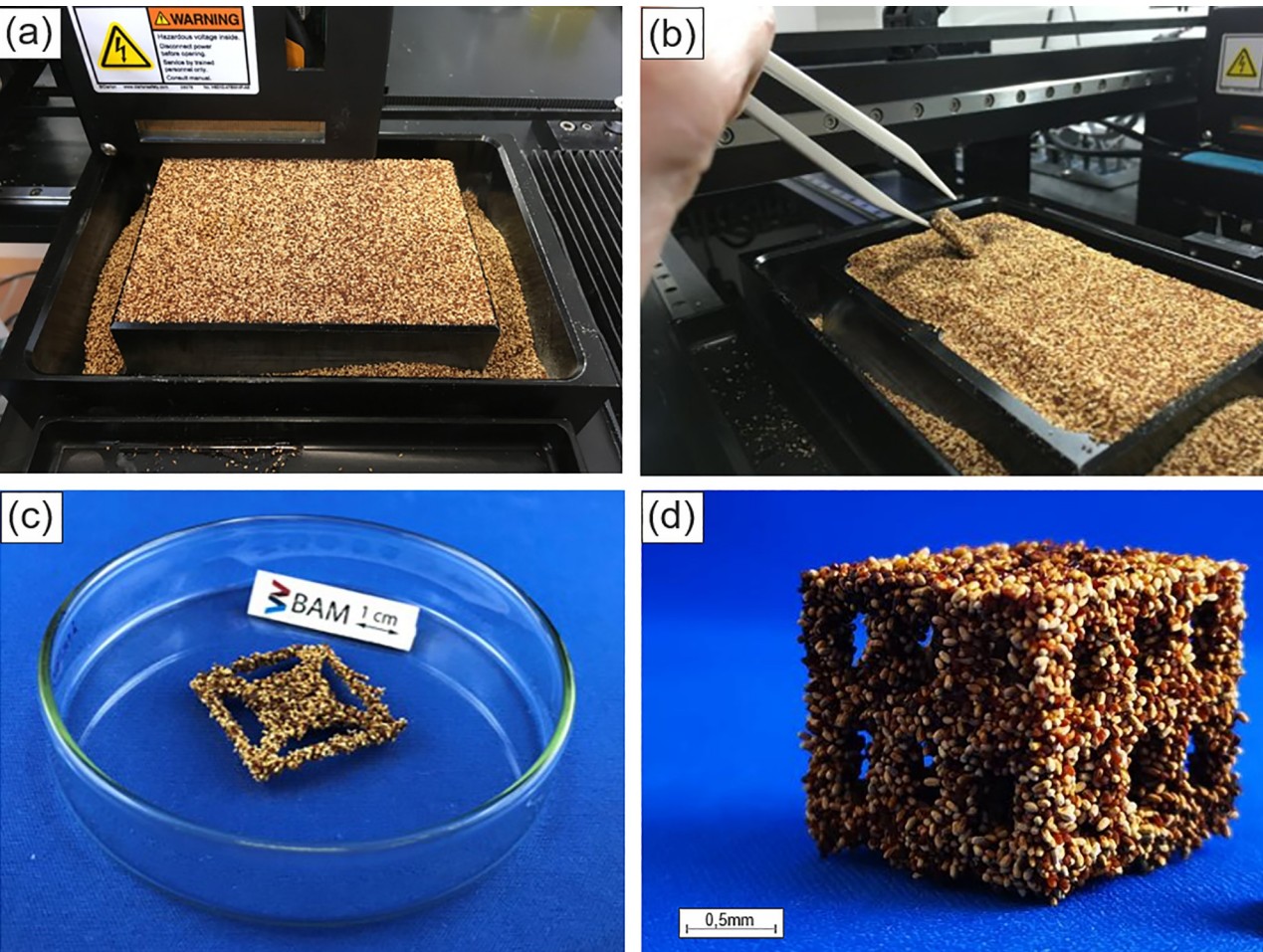

**Fig 7.** a) Binder jetting powder bed filled with drywood termite frass, b), c) part printed according to Fig 5B, d) printed part similar to the one shown in Fig 5A.

The frass pellets are approximately one order of magnitude larger than powders typically used in 3D printing. However, for printing wooden objects like furniture [16], coarse grained powders are beneficial for the deposition of thicker layers and higher building rates could be achieved. Structural details are reproduced on a scale of several millimeters but definitely not below one millimeter.

## Conclusions

Here we considered wood processed by the drywood termites Incisitermes marginipennis and the European house borer (EHB) Hylotrupes bajulus as feedstocks for 3D printing. This approach follows the general strategy of developing naturally available feedstocks as environmentally responsible substrates in material sciences. The quality of the powdery feedstocks, the so-called frass, provided by these insects during feeding in construction timber was very different in terms of processability in 3D printing. EHB frass reveals a flaky structure with poor packing density, whereas termite frass is consisting of pellets of almost uniform size and is packing very well. Despite of the different packing densities, both feedstocks could be spread out into thin homogeneous layers for the build-up of structures in the Binder jetting 3D printing process. At a size of 580 μm along the short- and 1050 μm at the long-axis, pellets of termite frass are approximately one order of magnitude larger than powders typically used in 3D printing. In printing wooden objects like furniture, coarse-grained powders allow the deposition of thicker layers and higher construction rates. The fine, sawdust-like frass particles of EHB, with their smaller particle size, are well suited to the printing of filigree structures. Both feedstocks do not qualify for printing of "ready to use" structures, as their mechanical strength is low, but printed structures are useful as preforms for further processing, such as infiltration, for improving mechanical strength. With their better flowability and packing density, this especially applies for structures printed from termite frass.

## Supporting information

**S1 Fig.**
(JPG)

**S2 Fig.**
(JPG)

**S3 Fig.**
(JPG)

**S4 Fig.**
(JPG)

**S5 Fig.**
(PNG)

**S6 Fig.**
(JPG)

**S7 Fig.**
(TIF)

**S8 Fig.**
(JPG)

**S9 Fig.**
(JPG)

**S1 File.**
(XLSX)

**S2 File.**
(XLS)

**S3 File.**
(XLSX)

## Author Contributions

**Conceptualization:** Rudy Plarre, Andrea Zocca, Sigrid Benemann, Anna A. Gorbushina, Yuexuan Li, Anja Waske, Janka Wilbig, Jens Günster.

**Data curation:** Andrea Spitzer, Alexander Funk.

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
