## [Decision Letter · Decision Letter 0]

22 Dec 2020

PONE-D-20-31824

Searching for biological feedstock material: 3D printing of wood particles from house borer and drywood termite frass

PLOS ONE

Dear Dr. Guenster,

Thank you for submitting your manuscript to PLOS ONE. After careful consideration, we feel that it has merit but does not fully meet PLOS ONE’s publication criteria as it currently stands. Therefore, we invite you to submit a revised version of the manuscript that addresses the points raised during the review process.

Your manuscript is accepted subject to minor changes as sought by the reviewers. Please submit a revised version accordingly.

We look forward to receiving your revised manuscript.

Kind regards,

Amitava Mukherjee, ME, Ph.D.

Academic Editor

PLOS ONE

Journal Requirements:

2.In your Data Availability statement, you have not specified where the minimal data set underlying the results described in your manuscript can be found. PLOS defines a study's minimal data set as the underlying data used to reach the conclusions drawn in the manuscript and any additional data required to replicate the reported study findings in their entirety. All PLOS journals require that the minimal data set be made fully available. For more information about our data policy, please see http://journals.plos.org/plosone/s/data-availability.

Reviewers' comments:

Reviewer's Responses to Questions

**Comments to the Author**

1. Is the manuscript technically sound, and do the data support the conclusions?

Reviewer #1: Yes

2. Has the statistical analysis been performed appropriately and rigorously? 

Reviewer #1: Yes

3. Have the authors made all data underlying the findings in their manuscript fully available?

Reviewer #1: Yes

4. Is the manuscript presented in an intelligible fashion and written in standard English?

Reviewer #1: Yes

5. Review Comments to the Author

Reviewer #1: The paper presents a novel approach use what is typically a waste from the wood industry and use Frass for 3D printing using binder jetting. The paper presents novel sustainability perspective and opens up avenues for impacting the bio product industry.

The paper can be accepted for publication, however the authors needs to address some of the following comments:

The experimental methodology for 3D printing needs be further detailed in the methodology section. Typically, the CAD models, print process parameters used for BJ need to be presented in the methods section. Currently, many are written in the results section.

Also, please include which instruments you used to measure tap and flow properties of the powder within the methods section.

As it stands there is intermixing of results and methods section and it goes back and forth within the results section. I understand that authors have used an iterative process of using smaller particles and larger particles to print two different geometry but putting the CAD models of these within the methods section will impart a flow that allows for the readers to know what the results section might have.

Also if in the results section, if authors could write the dimensional tolerance that they got from the printed parts. For eg. Fig. 5 and Fig 6 compares CAD and printed geometry. If the authors can just write how close was the printed pat when comparing it to Fig 5 and similarly for Fig 7 and Fig 8c within the text itself. Any measurement results will enable readers to understand with this novel process how well can you print your parts.

6. PLOS authors have the option to publish the peer review history of their article (what does this mean?). If published, this will include your full peer review and any attached files.

Reviewer #1: No

---

## [Author Response · Author response to Decision Letter 0]

17 Jan 2021

We would like to thank the reviewers for the constructive comments and have seriously worked over the manuscript. Actions undertaken in response to the points raised by the reviewers have been listed one be one in the following. 

Point by point response:

Point raised by the reviewer: The experimental methodology for 3D printing needs be further detailed in the methodology section. Typically, the CAD models, print process parameters used for BJ need to be presented in the methods section. Currently, many are written in the results section.

Action undertaken by the authors: We have moved all CAD drawings of parts printed and their introduction to the Materials and Methods section. 

Point raised by the reviewer: Also, please include which instruments you used to measure tap and flow properties of the powder within the methods section.

Action undertaken by the authors: We have added the following passage: “The tapped density was determined according to ISO 787-11 with a STAV 2003 type equipment from J. Engelsmann AG, Germany”

Point raised by the reviewer: As it stands there is intermixing of results and methods section and it goes back and forth within the results section. I understand that authors have used an iterative process of using smaller particles and larger particles to print two different geometry but putting the CAD models of these within the methods section will impart a flow that allows for the readers to know what the results section might have.

Action undertaken by the authors: Results and methods have been separated more consequently. In particular, the introduction of CAD models and printing parameters, as well as the introduction of µCT data have been moved from the results to the methods section.

Point raised by the reviewer: Also if in the results section, if authors could write the dimensional tolerance that they got from the printed parts. For eg. Fig. 5 and Fig 6 compares CAD and printed geometry. If the authors can just write how close was the printed pat when comparing it to Fig 5 and similarly for Fig 7 and Fig 8c within the text itself. Any measurement results will enable readers to understand with this novel process how well can you print your parts.

Action undertaken by the authors: Estimates of the dimensional tolerances of printed parts have been added in the results section.

---

## [Editor Report · Decision Letter 1]

21 Jan 2021

Searching for biological feedstock material: 3D printing of wood particles from house borer and drywood termite frass

PONE-D-20-31824R1

Dear Dr. Guenster,

We’re pleased to inform you that your manuscript has been judged scientifically suitable for publication and will be formally accepted for publication once it meets all outstanding technical requirements.

Kind regards,

Amitava Mukherjee, ME, Ph.D.

Academic Editor

PLOS ONE
---

## [Editor Report · Acceptance letter]

27 Jan 2021

PONE-D-20-31824R1 

Searching for biological feedstock material: 3D printing of wood particles from house borer and drywood termite frass 

Dear Dr. Günster:

I'm pleased to inform you that your manuscript has been deemed suitable for publication in PLOS ONE. Congratulations! Your manuscript is now with our production department. 

Kind regards, 

on behalf of

Professor Dr. Amitava Mukherjee 

Academic Editor

PLOS ONE